# Xylem Plasticity in *Pinus pinaster* and *Quercus ilex* Growing at Sites with Different Water Availability in the Mediterranean Region: Relations between Intra-Annual Density Fluctuations and Environmental Conditions

**Angela Balzano** [1], **Giovanna Battipaglia** [2], **Paolo Cherubini** [3,4] **and Veronica De Micco** [5,*]

1 Department of Wood Science and Technology, Biotechnical Faculty, University of Ljubljana, Jamnikarjeva 101, 1000 Ljubljana, Slovenia; Angela.Balzano@bf.uni-lj.si

2 Department of Environmental, Biological and Pharmaceutical Sciences and Technologies University of Campania "L. Vanvitelli", Via Vivaldi 43, I-81100 Caserta, Italy; giovanna.battipaglia@unicampania.it

3 Swiss Federal Research Institute WSL, Zürcherstrasse 111, CH-8903 Birmensdorf (Zurich), Switzerland; paolo.cherubini@wsl.ch

4 Department of Forest and Conservation Sciences, University of British Columbia, Forest Sciences Center, 2004-2424 Main Mall, Vancouver, BC V6T 1Z4, Canada

5 Department of Agricultural Sciences, University of Naples Federico II, Via Università 100, I-80055 Portici (Naples), Italy

* Correspondence: demicco@unina.it

**Abstract:** Fluctuations in climatic conditions during the growing season are recorded in Mediterranean tree-rings and often result in intra-annual density fluctuations (IADFs). Dendroecology and quantitative wood anatomy analyses were used to characterize the relations between the variability of IADF traits and climatic drivers in *Pinus pinaster* Aiton and *Quercus ilex* L. growing at sites with different water availability on the Elba island in Central Italy. Our results showed that both species present high xylem plasticity resulting in the formation of *L*-type IADFs (*L*-IADFs), consisting of earlywood-like cells in latewood. The occurrence of such IADFs was linked to rain events following periods of summer drought. The formation of *L*-IADFs in both species increased the hydraulic conductivity late in the growing season, due to their larger lumen area in comparison to "true latewood". The two species expressed greater similarity under arid conditions, as unfavorable climates constrained trait variation. Wood density, measured as the percentage of cell walls over total xylem area, IADF frequency, as well as conduit lumen area and vessel frequency, specifically in the hardwood species, proved to be efficient proxies to encode climate signals recorded in the xylem. The response of these anatomical traits to climatic variations was found to be species- and site-specific.

**Keywords:** false rings; Mediterranean ecosystems; oak; pine; tree-rings; wood anatomical traits

## 1. Introduction

Mediterranean ecosystems are characterized by remarkable seasonal fluctuations in water availability during the growing season, which often lead to pronounced drought stress in summer. Intra-seasonal climatic fluctuations are becoming more and more marked as a consequence of ongoing climate change (IPCC, 2018). Therefore, an increase in the frequency of severe summer drought and heatwave episodes will likely affect woody plant growth and the productivity of forests [1–3].

The increased efficiency of the plant hydraulic system is fundamental for plant survival in Mediterranean environments. In these environments, plants must efficiently transport water when it is available (through large earlywood conduits) while maintaining a low propensity for embolism (through narrow latewood conduits) when water is scarce [4–6]. Functional traits in the wood of Mediterranean species are the result of particular patterns of xylogenesis, allowing cambial cells to produce more than two bands of alternating earlywood and latewood during the calendar year [6–10]. The number of conductive elements produced by the cambium, their lumen size and cell wall thickness are all controlled by both physiological processes and environmental conditions [11–14] and can affect the functionality of the xylem. Fluctuations in climatic conditions during the growing season are thus recorded in xylem and often result in wood anatomical "anomalies", changing the aspect of the typical tree-rings formed in temperate regions [15]. Such anomalies, called intra-annual density fluctuations (IADFs), have recently been classified as functional wood traits, as they play a functional role in the tree hydraulic system [15,16]. IADFs are regions within a tree-ring where abrupt changes in density occur as a results of the alternation of different layers of earlywood (large conduits with thin cell walls) and latewood (narrow conduits with thick cell walls) characterized by different hydraulic conductivities [15–17]. Different types of IADFs have been detected in Mediterranean species and their position within the ring is determined by the time of occurrence of the triggering factor [18]. The most frequent types of IADFs occurring in tree-rings of Mediterranean species are *L*-IADFs and *E*-IADFs [15]. *L*-IADFs appear as earlywood-like cells in the latewood, and their occurrence has been linked with the restoration of favorable conditions of water availability due to rain events following conditions of severe drought stress [18–21]. *E*-IADFs, conversely, appear as latewood-like cells in earlywood, and their formation has been hypothesized to be due to stomatal closure under summer drought stress to reduce the embolism risk [13,22]. Several studies have demonstrated the dependency of IADF frequency on species, cambial age, climatic and microclimatic conditions [9,17,19,23–29].

Given that wood anatomy and plant hydraulics play a key role in understanding species-specific responses and the ability to cope with rapid environmental changes [30,31], an understanding of the ecological meaning of IADFs is crucial to comprehend the adaptive capability of plants, based on xylem plasticity [32]. Indeed, to understand the ecological function of IADFs as an acclimation response to climate, studies of intra-annual xylem plasticity across a range of species growing in a range of environmental conditions are required. The obtained information would allow us to predict possible future ecosystem dynamics. The analysis of the quantitative anatomical traits of IADFs can help us obtain information about the relationships between environmental factors and tree growth at the intra-seasonal level [19,33–35]. However, the quantification of xylem cell traits (e.g., lumen area, wood density, conduit frequency) is time-consuming and represents a constraint, although different semi-automatic approaches have been developed [31,36]. The tracheidogram method has been developed to identify the relationships between environmental conditions and growth in conifers, based on the lumen diameter and cell wall thickness of tracheids [11]. In hardwoods, the study of specific regions within tree-rings has been used to evaluate the influence of environmental conditions on vessel lumen area throughout the growing season [19,34,37,38]. Recently, the concept of tracheidogram has also been applied to analyse vessels in hardwoods which allows us to link their features to climate [35,39–41].

However, the physiological mechanism triggering IADF formation, and how tree-ring anatomy responds to climate, has not been completely unrevealed yet, mainly due to the high variability in their occurrence and their dependency on many factors, both intrinsic and external [16,32].

In this study, we analysed the xylem plasticity of two of the most widespread evergreen tree species in the Mediterranean area, *Pinus pinaster* Aiton and *Quercus ilex* L., which are species prone to produce IADFs [13,33,42–46]. The tree-ring series of the two species were analysed from trees growing at two sites with contrasting soil water availability, namely a mesic (wet) and a xeric site (dry).

Our specific aims were: (1) to verify the aptitude of the oak and pine species to form IADFs in different environmental conditions; (2) to classify IADFs in both species to verify whether water

availability at the two sites affected IADF frequency and type; (3) to link intra-annual variation of anatomical traits with climate parameters at intra-seasonal scale. To achieve our goals, we used a multidisciplinary approach combining dendrochronology, quantitative wood anatomy and climatic parameters. In particular, we used the approach of highlighting intra-IADF variability through the measurement of vessel size data in continuum within the tree-ring [27,28,35,41]. We also quantified other wood traits, namely wood density, in both species, and vessel frequency, in *Q. ilex*, along tree-ring width. We assessed the potentiality of the different traits to be used as indicators of intra-seasonal variations of climate parameters.

## 2. Materials and Methods

### 2.1. Species and Study Site

The study was conducted on *Pinus pinaster* Aiton and *Quercus ilex* L. trees growing on Elba Island in the Tyrrhenian sea (Central Italy). The site was characterized by a Mediterranean climate, with a mean annual temperature of 15.41 °C and precipitation mainly concentrated in autumn and winter, with an average of 417 mm during the period 1960–2007 (Figure 1). We selected two sampling sites with different amounts of soil moisture: the first, xeric (dry), located on Monte Perone (42°46′ N, 10°12′ E, 420 m a.s.l.), presented xeric species and shrubs, and a soil water holding capacity reduced by more than 40% compared to the second, the mesic (wet) site, located in the Nivera Valley (42°46′N, 10°11 E, 460 m a.s.l.), presenting more mesic species [19]. Precipitation and temperature data were obtained from the Portoferraio meteorological station located at 10 km from the sites (42°49′N, 10°20′ E, 25 m asl). Details on site characteristics are given in [19].

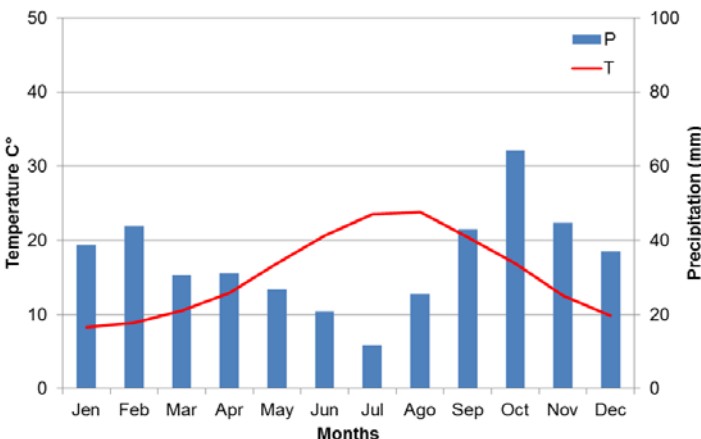

**Figure 1.** Mean monthly average temperature and mean total monthly precipitation recorded in Elba Island during the period 1960–2007.

### 2.2. Tree-ring Data and IADF Frequency

We selected 10 trees per species at each site, sampling two cores at breast height per each plant, from East and West directions, with a Pressler increment borer (0.5 cm diameter). The chronologies obtained from individual cores where averaged. Since our sites were placed in the National Park of the Tuscan Archipelago, we were allowed to sample only a minimum number of specimens. However, we could obtain reliable chronologies, assessing the strength of the chronologies by the 'expressed population signal' (EPS threshold of 0.85) [47]. The cores were dried and polished to obtain visual cross-dating [48] and to allow the identification and classification of IADFs according to [15]. Tree-ring width (TRW) was then measured using a LINTAB™ measurement table (Frank Rinn, Heidelberg, Germany), with a resolution of 0.01 mm and individual series were cross-dated and detrended (20-year spline), to remove non-climatic signals using the Dendrochronology Program Library in R [49]. The Gleichläufigkeit (GLK- a measure of the year-to-year agreement between the interval trend of two chronologies based

upon the sign of agreement) and rbar (which is a measure of the common variance between the single series in a chronology and series intercorrelation) were calculated [48].

We detected *L*-IADFs in the tree-ring series of both *P. pinaster* (Figure 2a,b) and *Q. ilex* (Figure 2c,d). Their relative frequency (IADFs yr–1) was calculated as the number of trees that presented IADFs in a given year, divided by the total number of sampled trees in that year. Stabilized IADF frequency was calculated according to [50] as f = Fn 0,5. The frequency was calculated on ten cores for each species.

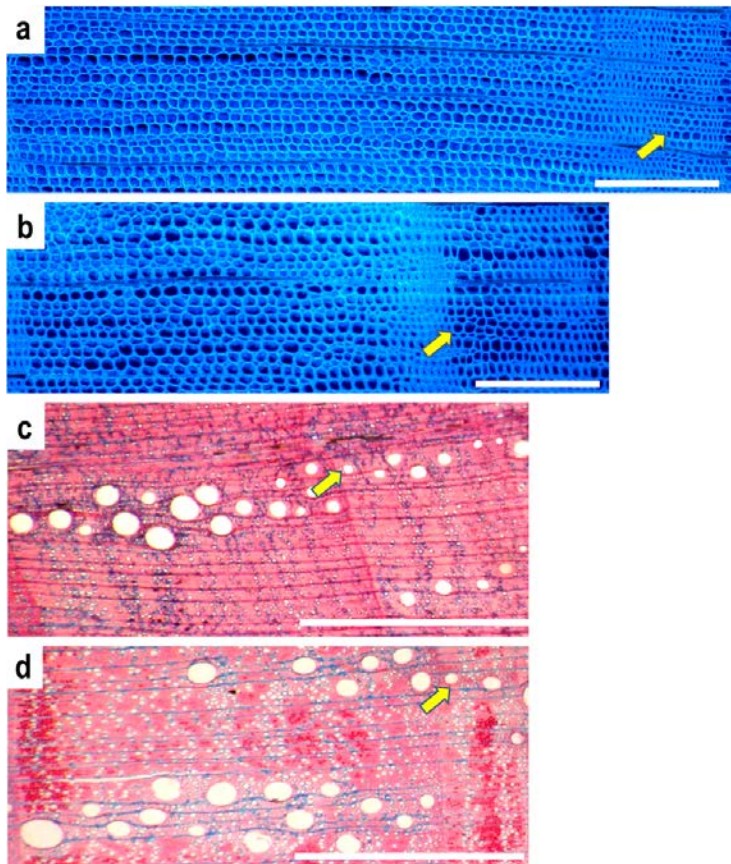

**Figure 2.** Tree-ring with *L*-type intra-annual density fluctuations (*L*-IADFs) at dry site (**a**) and wet site (**b**) in *Pinus pinaster* and tree-rings with *L*-IADFs at dry site (**c**) and wet site (**d**) in *Quercus ilex*. Yellow arrows indicate the beginning of earlywood-like cells in the latewood. Ba*r* = 500 μm.

*2.3. Microscopy and Quantitative Wood Anatomy*

For *P. pinaster*, the cores were directly observed under an epi-fluorescence microscope (BX60, Olympus), equipped with a Mercury lamp, bandpass filter 330–385 nm, dichromatic mirror 400 nm and above and barrier filter 420 nm and above. Such settings allow the maximisation the contrast between lignified cell walls and tracheid lumen [51]. Digital images were captured with the Olympus CAMEDIA C4040 camera.

For *Q. ilex*, cross microsections of cores (15μm thick) were obtained with a sliding microtome. Sections were stained with a safranin (0.04 %) and astra blue (0.15 %) water solution [52], and mounted in Euparal (Bioquip Rancho Dominguez, California). After this, they were analysed under a light microscope (Olympus BX60, Hamburg, Germany), and digital images were captured as reported for *P. pinaster*.

We considered five plants per species at both sites; in particular, we analyzed tree-rings corresponding to the following periods: (a) from 1969 to 2007 in *P. pinaster*; (b) from 1990 to 2007 in *Q. ilex*. The variability of the anatomical features along the width of each ring was quantified with digital image analysis through the image analysis software Analysis 3.2 (Olympus) which provides

image acquisition and quantitative measurements of cytological and anatomical traits. Lumen area of vessels of *Q. ilex* and tracheids of *P. pinaster* were semi-automatically quantified in all elements encountered, moving from the beginning (earlywood) to the end (latewood) of each tree-ring in chosen horizontal transects (radial lines of tracheids in *P. pinaster* and vessels in a radial bundle in *Q.ilex*), following the standardized progressive number method described by [41]. In brief, during the measurement, the progressive number of each element was recorded, moving from earlywood to latewood, to build dispersion graphs with Y and X as coordinates of each conductive element to visually compare rings with and without IADFs at the wet and the dry site. The progressive number of vessels within the tree-ring was standardized dividing it by the total number of vessels encountered along the transect and multiplying by 100, obtaining series of vessels/tracheids size by the same principle of tree-ring width chronologies. For each of the considered parameters, a dispersion graph was drawn, where each conduit was characterised by two coordinates: Y, corresponding to the measured lumen parameter and X, corresponding to the distance from the beginning of the ring, expressed as a percentage of the total ring width. The patterns of vessels/tracheids size variability along ring width were visually compared in rings with and without IADFs. In the standardized data series, we calculated interpolation equations (fourth-order polynomial curve) with confidence intervals using the option non-linear curve fit into the R system [53].

Once we obtained the series of tracheid/vessel lumen areas, the tree-ring without IADFs were partitioned into 4 regions (each region corresponding to the 25% of total ring width: Region 1 and 2 coinciding with earlywood and Region 3 and 4 coinciding with latewood), while the tree-ring with IADFs were partitioned into 5 regions (each region corresponding to the 20% of total ring width: Region 1 and 2 coinciding with earlywood, Region 3 and 4 coinciding with latewood and Region 5 coinciding with the IADF) along the radial direction from the beginning (Region 1) to the end (Region 4 for tree-rings without IADFs; Region 5 for tree-rings with IADFs). We considered the mean value of vessels/tracheids lumen area per each region of the ring to perform climate correlations with intra-annual resolution.

Furthermore, we analysed wood density (measured as the percentage of cell walls over total xylem area) for both species. In *Q. ilex*, we calculated vessel frequency (the number of vessels per $mm^2$, determined by counting the vessels present in a known area, according to Wheeler et al. (1989), in the same tree-rings considered for measurements of tracheid/vessel lumen area. To compare mean values of anatomical traits between different sectors of tree-rings, each tree-ring without IADFs was partitioned in the above-reported 4 regions, while for tree-rings with IADFs, a sort of standardization of the conduit position was performed by making the 80% of the width of the tree-ring with IADFs coinciding with 100% of the width of tree-ring without IADFs. In detail, tree-ring width in the presence of IADFs were divided into the above-reported 5 regions: the first four regions were considered as coinciding with those of rings without IADFs, while the fifth region was considered as an additional growth layer containing the IADF. The distinction between earlywood and latewood was based on the application of Mork's definition [54].

The stabilized IADF frequency and all anatomical parameters were subjected to Shapiro–Wilk and Kolmogorov–Smirnov tests to check for normality. All data per each species and site were processed with one-way analysis of variance (ANOVA) to compare different tree-ring regions, using Student–Newman–Keuls coefficient for multiple comparison tests ($p < 0.05$). The SPSS statistical package was used (SPSS Inc.; Chicago, IL, USA).

*2.4. Climate Signal*

We correlated measured anatomical traits and stabilized IADF frequency to meteorological data by computing Pearson coefficients ($p < 0.05$) considering the following periods: (a) 1969–2007 for *P. pinaster* at both sites; (b) 1994–2007 at dry site and 1975–2007 at the wet site for *Q. ilex*. We considered the mean value of vessel/tracheid lumen area, wood density and vessel frequency in different regions within the tree-ring to obtain climate correlation with intra-annual resolution. Precipitation and maximum

temperature at three-months scale resolution, from September of the previous year to December of the current year, were used in the analysis. All correlations were calculated using the treeclim R package [55].

## 3. Results

### 3.1. Tree-Ring Dating and IADFs Occurrence

The mean chronologies of *P. pinaster* covered the period from 1964 to 2007 (mean age 38 years, GLK 0,65, rbar 0,47, TRW 2.06 ± 1.64 mm) for the trees at wet site (Figure 3a), and from 1960 to 2007 (mean age 32 years, GLK 0,55, rbar 0,29, TRW 5.17 ± 2.14 mm) for the trees at the dry site (Figure 3b). For *Q. ilex* the total time span of the chronologies extended from 1994 to 2007 (mean age 11, GLK 0,65, rbar 0,39, TRW 4.19 ± 2.09 mm) for the trees at the wet site (Figure 3c), and from 1975 to 2007 (mean age 24, GLK 0,59, rbar 0,17, TRW 1.72 ± 0.69 mm) for those at the dry site (Figure 3d).

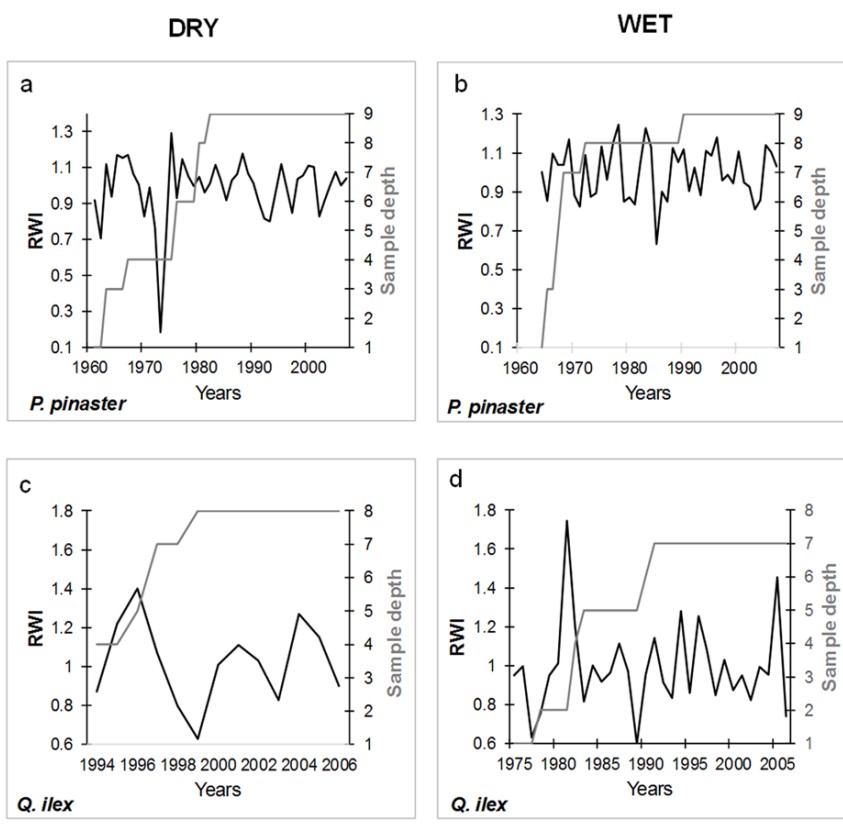

**Figure 3.** Average ring width chronologies (Black lines, ring width index RWI) after detrending in *P. pinaster* at wet site (**a**) and at dry site (**b**), and in *Q.ilex* at wet site (**c**) and dry site (**d**). Grey lines shows the sample depth (number of samples for each year).

*P. pinaster* showed a higher frequency of IADFs than *Q. ilex* at both sites. In both species, IADF frequency varied according to the site. Indeed, in the case of *Q. ilex* at the dry site the occurrence of IADFs was higher (*f* = 1.0) than at the wet site (*f* = 0.5). Conversely, *P. pinaster* trees at the wet site appeared to be more prone to form IADFs (*f* = 2.1) than those at the dry site (*f* = 1.2).

### 3.2. Xylem Traits

In *P. pinaster*, the series of lumen areas of tracheids obtained using the standardized progressive number method essentially showed a common pattern at both sites, being clearly bimodal in the rings with IADFs. The lumen areas of tracheids in rings without IADFs decreased from earlywood to latewood (Figure 4a,b blue line), while in the rings with IADFs it increased around the 70% of

vessels along the tree-ring width (Figure 4a,b red line), where sometimes earlywood-like cells were present. Tracheid lumen area values in rings with IADFs at the wet site (Figure 4a, red line) increased more markedly at the end of the ring than in those at the dry site (Figure 4b, red line) ($r^2 = 0.362$ for rings with IADFs, $r^2 = 0.410$ for rings without IADFs at the wet site; $r^2 = 0.497$ for rings with IADFs, $r^2 = 0.596$ for rings without IADFs at the dry site).

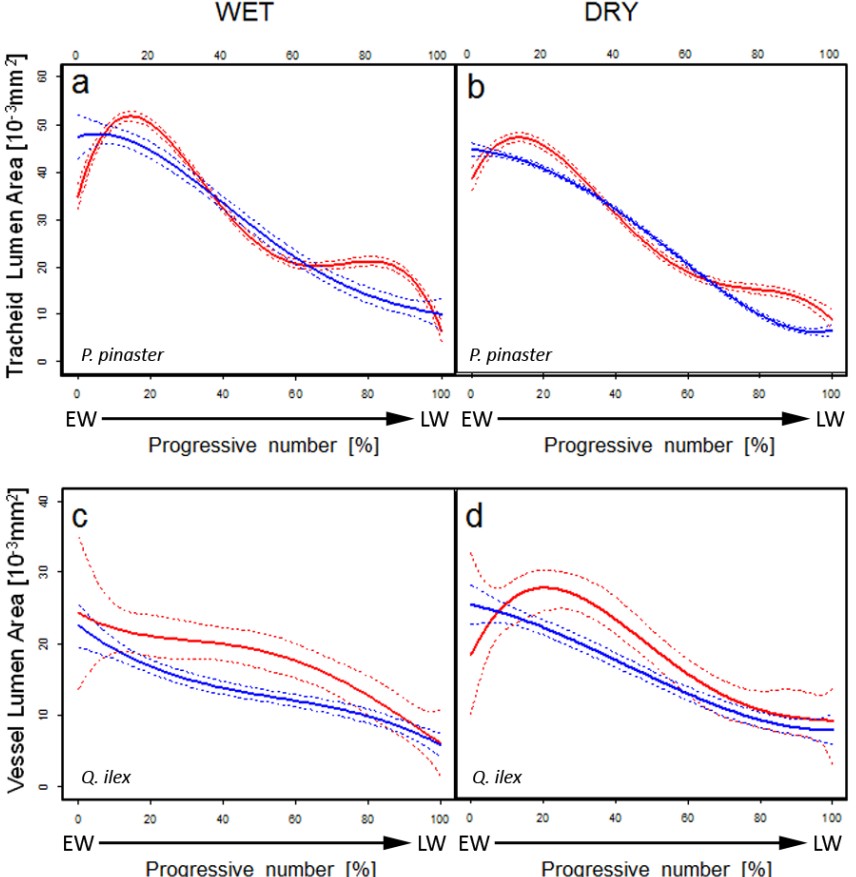

**Figure 4.** Variation in tracheid/vessel size along ring width (from earlywood—EW to latewood—LW) shown by plotting the set of the tracheid/vessel-lumen-area standardized data of rings with intra-annual density fluctuations (IADFs) (red line) and without IADFs (blue line) for *P. pinaster* at wet (**a**) and dry site (**b**) and *Q. ilex* at wet (**c**) and dry site (**d**). Confidence intervals are shown as dashed lines.

Wood density measurements showed an opposite pattern compared to tracheid lumen area. In the rings without IADFs (Figure 5a,b), at both sites, wood density gradually increased until it reached the maximum values at the end of the ring. In rings with IADFs, density decreased at the end of the ring instead, particularly more markedly at the wet site (Figure 5c) than at the dry site (Figure 5d).

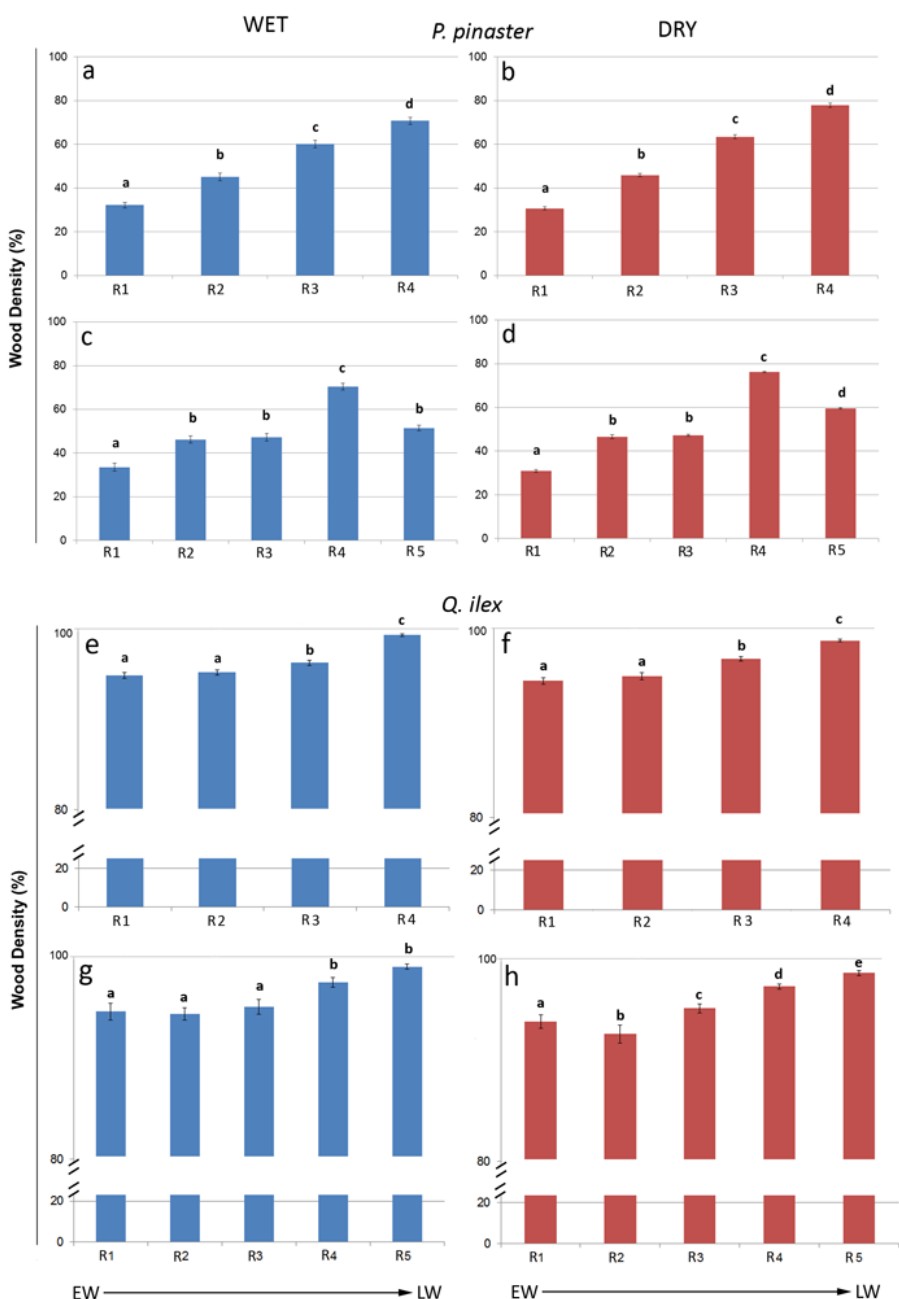

**Figure 5.** Wood density (measured as percentage of cell walls over total xylem area) in tree-rings of *P pinaster* without (**a,b**) and with (c,d) IADFs at the wet (**a,c**) and dry sites (**b,d**), and of *Q. ilex* without (**e,f**) and with (**g,h**) IADFs at the wet (**e,g**) and dry sites (**f,h**). Mean values and standard errors are shown. Different letters indicate significantly different values ($p < 0.05$) between different regions (R1–R4 and R1–R5 for tree-rings without and with IADFs respectively) oriented from earlywood (EW) to latewood (LW).

In *Q. ilex*, the trends of lumen area vessel also showed differences between rings with and without IADFs at the two sites. At the wet site, the vessel lumen area at the beginning of ring without IADFs (Figure 4c, blue line) showed higher values than those in the ring with IADFs (Figure 4c, blue line). At the dry site, the values of vessel lumen area in rings with IADFs (Figure 4d, red line) showed an increase at the end of the ring (90%). In rings without IADFs (Figure 4d, blue line), the vessel lumen area decreased gradually from the beginning to the end of the ring. Rings with IADFs at the dry site (Figure 4d, red line) showed a peak in the values of lumen area at the end of the ring that was not

present in rings without IADFs at the wet site (Figure 4c, blue line) ($r^2 = 0.211$ for rings with IADFs, $r^2 = 0.163$ for rings without IADFs at the wet site; $r^2 = 0.258$ for rings with IADFs, $r^2 = 0.226$ for rings without at the dry site).

In both *P. pinaster* and *Q.ilex*, at both sites, density increased gradually from the beginning to the end of the ring when the IADFs were not present (Figure 5e,f). However, in contrast to *P. pinaster*, in the ring with IADFs of *Q.ilex*, wood density did not decrease at the end of the ring (Figure 5g,h).

Vessel frequency showed higher values at the wet site (Figure 6a,c) than at the dry site (Figure 6b,d), and in general the highest values were detected in the middle of the ring. Vessel frequency decreased always at the end of the ring but more markedly in rings with IADFs (Figure 6c,d).

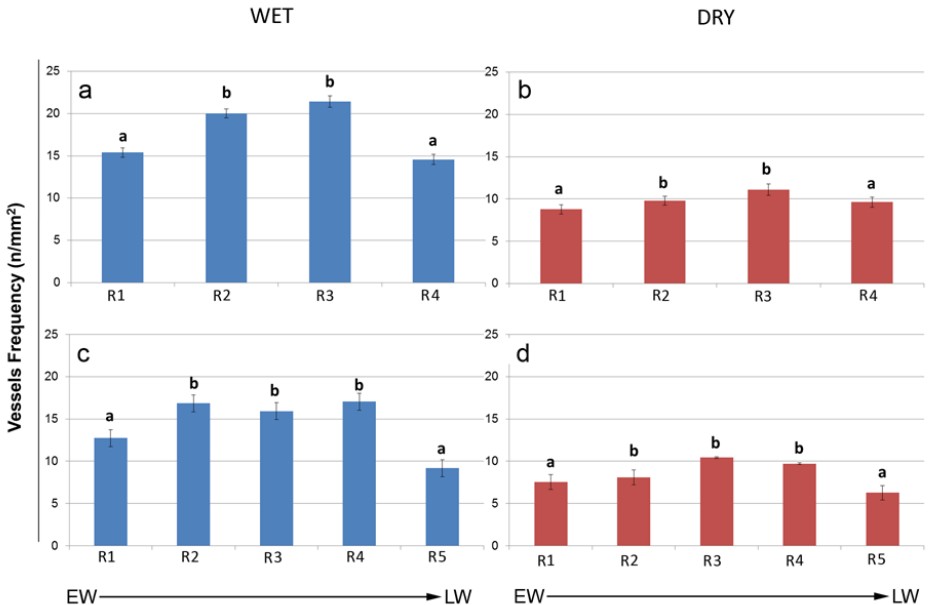

**Figure 6.** Vessel frequency in *Q. ilex* tree-rings without (**a**,**b**) and with (**c**,**d**) IADFs at the wet (**a**,**c**) and dry sites (**c**,**d**). Mean values and standard errors are shown. Different letters indicate significantly different values ($p < 0.05$) between different regions (R1–R4 and R1–R5 for tree-rings without and with IADFs respectively) oriented from earlywood (EW) to latewood (LW).

*3.3. Climate Correlations*

Correlations between anatomical traits and climate data showed that, in *P. pinaster* growing at the wet site, there was a negative relation between lumen area of latewood tracheids and precipitation from October to December of the previous year (Figure 7a, Region 4, $r = -0.387$, $p < 0.05$), while a positive relation occurred between the precipitation from the current-year October to December and the tracheid lumen area in the region of IADF (Figure 7a, Region 5, $r = 0.437$, $p < 0.05$). A negative correlation was found between earlywood tracheid lumen area and maximum temperature from October to December of the previous year (Figure 7b, Region 2, $r = -0.352$, $p < 0.05$). At the dry site, there were no significant correlations between tracheid lumen area and precipitation (Figure 7c), while maximum temperature from April to June was positively correlated with earlywood tracheid lumen area (Figure 7d, Region 1, $r = 0.432$, $p < 0.05$). At the wet site, there were no significant correlations between wood density and precipitation (Figure 7e), while a positive correlation between wood density in earlywood and maximum temperature from January to March was found (Figure 7f, Region 2, $r = 0.396$, $p < 0.05$). At the dry site, wood density in earlywood was positively correlated with precipitation from January to March (Figure 7g, Region 2, $r = 0.325$, $p < 0.05$), while negatively correlated with maximum temperature from December to October of the previous year (Figure 7h, Region 2, $r = -0.398$, $p < 0.05$). Latewood density was negatively related to maximum temperature from July to September (Figure 7h, Region 3, $r = -0.269$, $p < 0.05$).

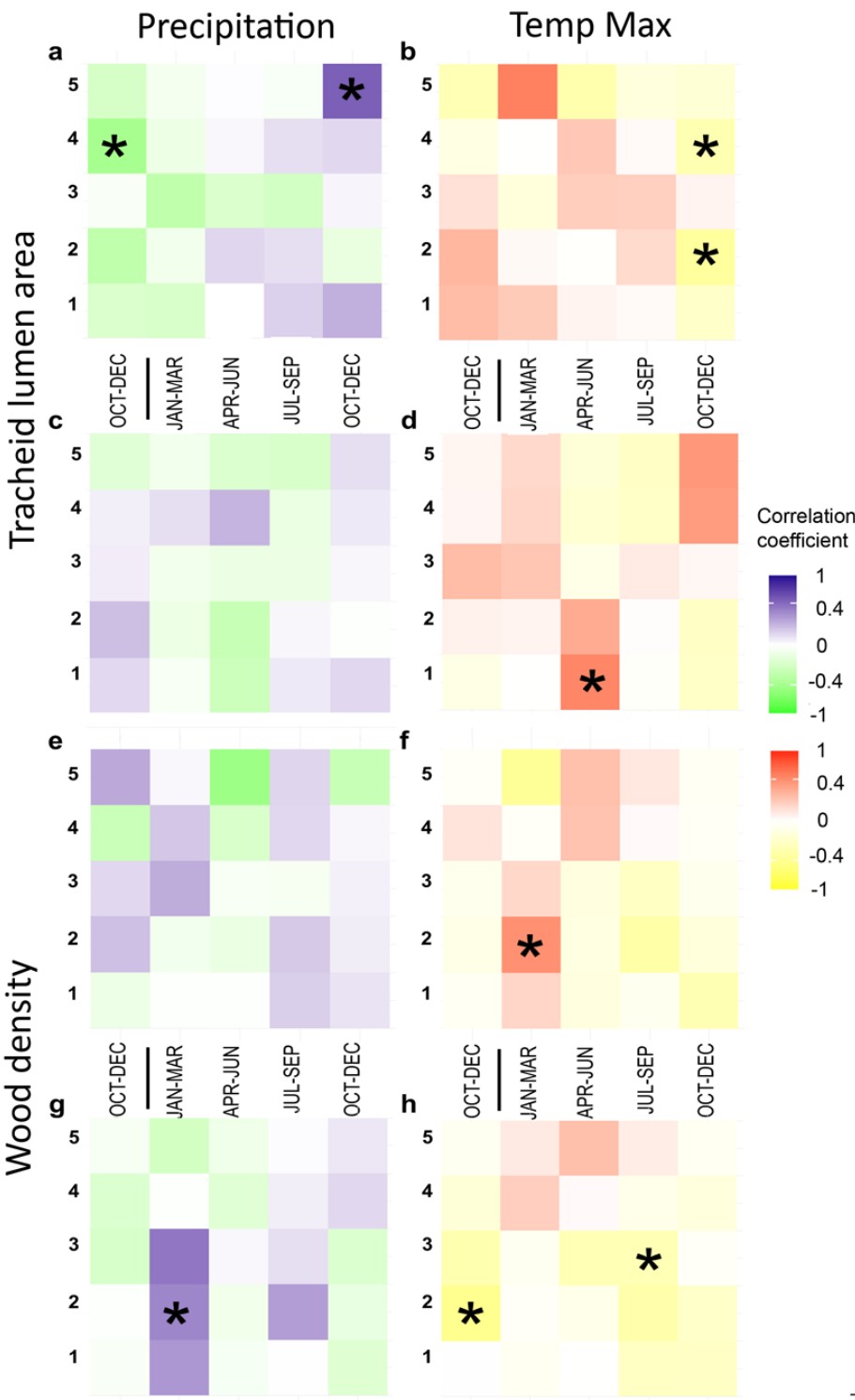

**Figure 7.** Climate–growth associations computed by comparing tracheid lumen area (**a**–**d**) and wood density (**e**–**h**) with precipitation (**a**,**c**,**e**,**g**) and maximum temperature (**b**,**d**,**f**,**h**) in *P. pinaster* at the wet (**a**,**b**,**e**,**f**) and the dry site (**c**,**d**,**g**,**h**). Correlations were calculated from October of the previous year to December of the current year of tree-ring formation (x axes) and partitioning the tree-ring without IADFs in four regions and with IADFs in five regions along the radial direction from the beginning (Region 1) to the end (Region 4–Region 5) of the ring (y axes). Significant correlations (*p* < 0.05) are indicated with asterisks.

In the *Q. ilex* growing at wet site, we found a positive correlation between the lumen area of latewood vessels and precipitation in the period from October to December of the previous year (Figure 8a, Region 4, $r = 0.256$, $p < 0.05$), and a positive correlation between vessel lumen area of IADF region and precipitation from October to December of the current year (Figure 8a, Region 5, $r = 0.311$, $p < 0.05$). No significant correlations were found between the vessel lumen area and maximum temperature at this site (Figure 8b). At the dry site, vessel lumen area of latewood was negatively correlated with precipitation and positively correlated with maximum temperature in the period from April to June (Figure 8c,d, Region 4, $r = -0.567$, $p < 0.05$). At the wet site, latewood density was negatively correlated with precipitation in the period from October to December (Figure 8e, Region 4, $r = -0.438$, $p < 0.05$), while precipitation during these months was positively correlated with wood density in the IADF region (Figure 8e, Region 5, $r = 0.308$, $p < 0.05$). The maximum temperature in the same months showed a positive correlation with wood density at the end of earlywood and beginning of latewood (Figure 8f, Region 2, $r = 0.553$, $p < 0.05$, and Region 3, $r = 0.439$, $p < 0.05$). At the dry site, latewood density was negatively correlated with precipitation in the period from April to June (Figure 8g, Region 3, $r = -0.554$, $p < 0.05$), while wood density in the region of IADF was negatively correlated with maximum temperature in the period from January to March (Figure 8h, Region 5, $r = -0.652$, $p < 0.05$). At the wet site, we found a positive correlation between vessel frequency in earlywood and precipitation in the period from July to September (Figure 8i, Region 1, $r = 0.413$, $p < 0.05$). Vessel frequency at the beginning of latewood was negatively correlated with maximum temperature in the period from January to March (Figure 8l, Region 3, $r = -0.398$, $p < 0.05$), while vessel frequency in the second part of the latewood was negatively correlated with maximum temperature in the period from October to December (Figure 8l, Region 4, $r = -0.247$, $p < 0.05$). At the dry site, vessel frequency in the earlywood and at the beginning of latewood was positively correlated with precipitation in the period from April to June (Figure 8m, Region 1, $r = 0.618$, $p < 0.05$, and Region 3, $r = 0.522$, $p < 0.05$). Vessel frequency in the IADF region was positively correlated with the maximum temperature in the above-mentioned months (Figure 8n, Region 5, $r = 0.347$, $p < 0.05$).

Concerning the correlation between *L*-IADFs stabilized frequency and climate data, we found that the *P. pinaster* at the wet site's frequency was negatively correlated to precipitation in November of the previous year (Figure 9a) and positively correlated with the maximum temperature in September (Figure 9b). Instead, at the dry site, the occurrence of *L*-IADFs was positively correlated with the maximum temperature in December (Figure 9c). In *Q. ilex* growing at the wet site, the *L*-IADF frequency was negatively correlated with maximum temperature of the previous September and current November (Figure 9d). At the dry site, there were no significant correlations between *L*-IADF frequency and data of precipitation and maximum temperature.

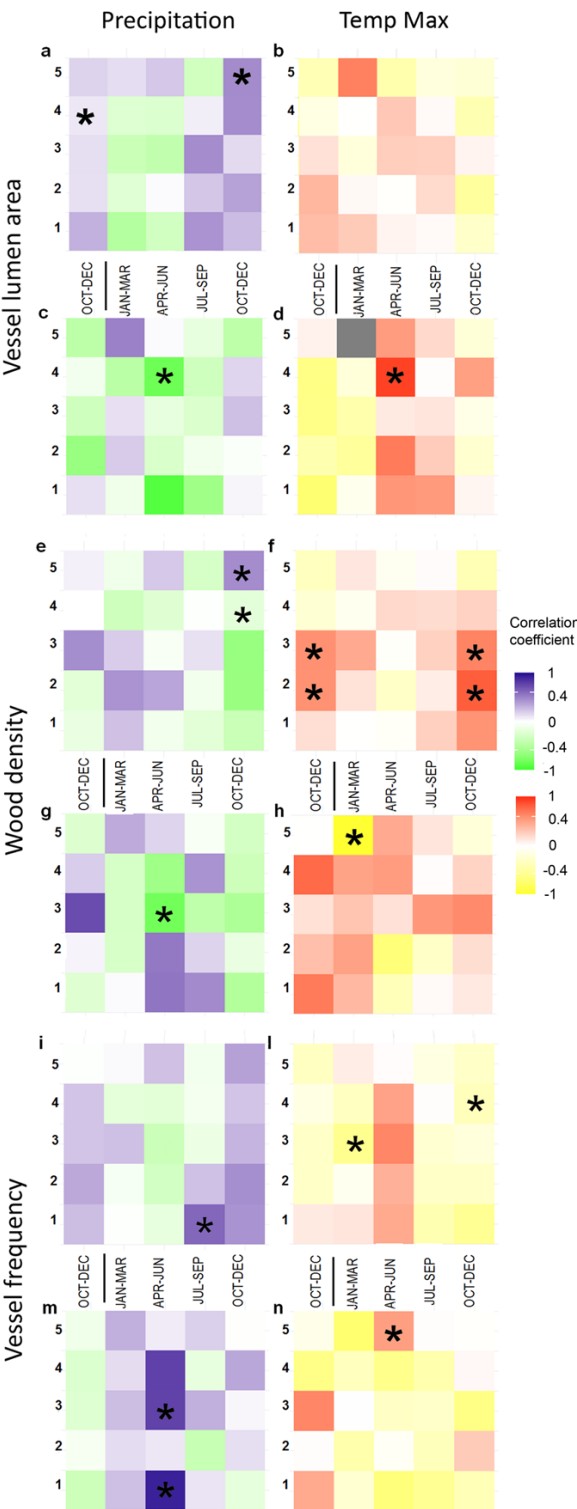

**Figure 8.** Climate–growth associations computed by comparing vessel lumen area (**a**–**d**), wood density (**e**–**h**) and vessels frequency (**i**,**l**,**m**,**n**) with precipitation (**a**,**c**,**e**,**g**,**i**,**m**) and maximum temperature (**b**,**d**,**f**,**h**,**l**,**n**) in *Q. ilex* in wet (**a**,**b**,**e**,**f**,**i**,**l**) and in dry site (**c**,**d**,**g**,**h**,**m**,**n**). Correlations were calculated from October of the previous year to December of the current year of tree-ring formation (x axes) and partitioning the tree-ring without IADFs in four regions and with IADFs in five regions along the radial direction from the beginning (Region 1) to the end (Region4–Region 5) of the ring (y axes). Significant correlations ($p < 0.05$) are indicated with asterisks.

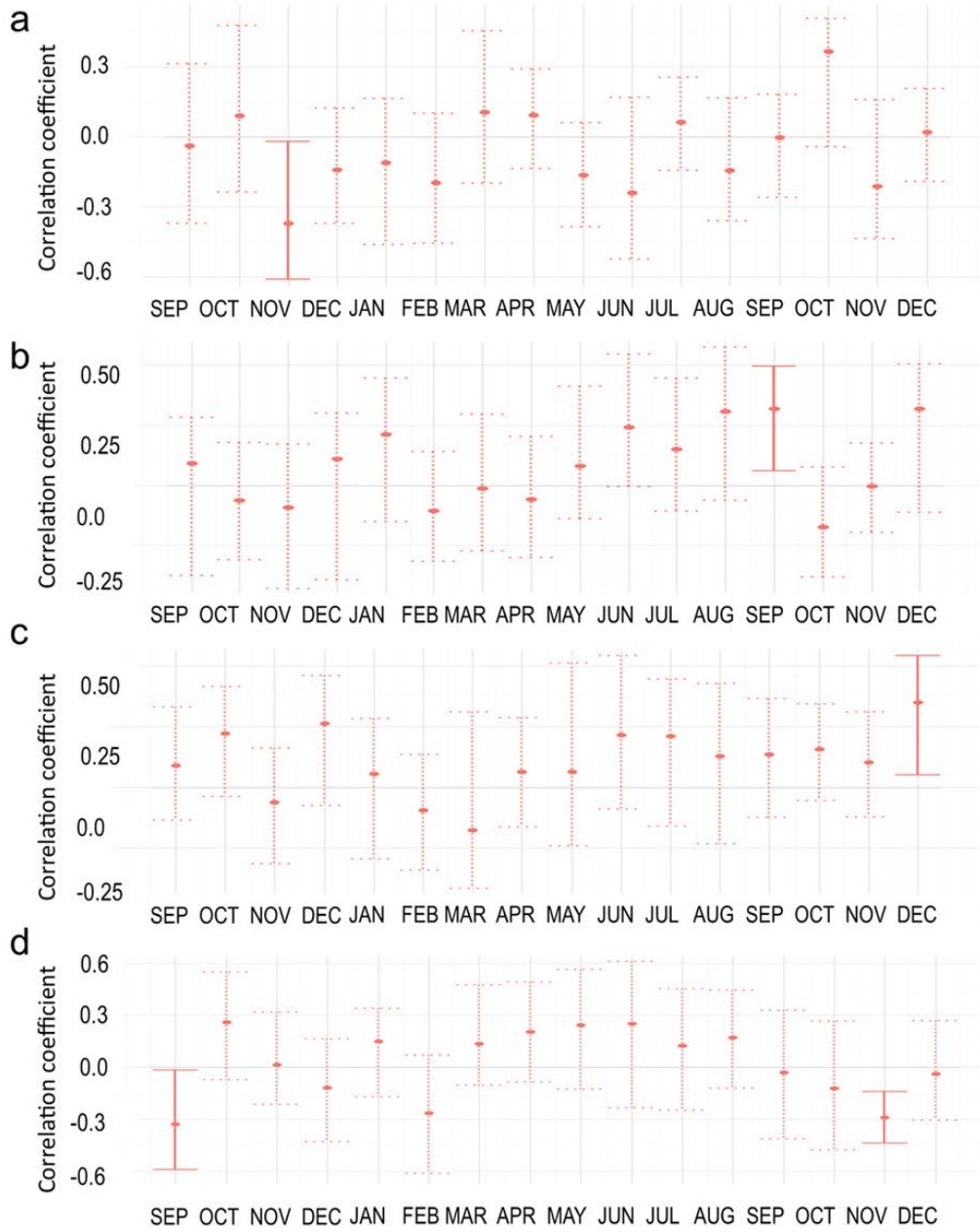

**Figure 9.** Climate–IADFs associations computed by comparing *L*-IADFs frequency in *P. pinaster* (**a**–**c**) and *Q. ilex* (**d**) with precipitation (**a**) and maximum temperature (**b**–**d**) in wet (**a**,**b**,**d**) and dry site (**c**). Correlations were calculated from September of the previous year to December of the current year of tree-ring formation (x axes). Significant correlations ($p < 0.05$) are indicated with red lines.

## 4. Discussion

Wood anatomical features, such as vessel (or tracheid) lumen area and vessel density, measured in long tree-ring series can be used as climatic proxies with high temporal resolution [39,56,57], thus improving the information traditionally gained from tree-ring width [31,33,58]. In our study, the collection of conduit size data in continuum within the tree-ring chronologies [35,41] and the quantification of wood density and vessel frequency in specific regions within the tree-rings showed the patterns of xylem variability of *P. pinaster* and *Q. ilex* in mesic (wet) and xeric (dry) conditions. We identified IADFs in both species, classifying most of them as *L*-type according to [15].

### 4.1. The Role of Water Availability in P. pinaster

The *Pinus pinaster* series of tracheid lumen areas showed a clear bimodal pattern due to the occurrence of *L*-IADFs already identified in this species growing at other Mediterranean sites [44,45,59, 60], with a decrease in cambial activity in summer and winter [61].

The difference in water availability at the two sites seemed to influence the plasticity of xylem that responded in different ways to the variations in climatic conditions, as also found in other species growing at the same sites [19,22].

The tree-rings of *P. pinaster* growing at the wet site appeared to be more prone to form *L*-IADFs than those at the dry site. At both sites, tree-rings with IADFs showed an increase in tracheid lumen area combined with a decrease in wood density at the end of tree-ring width. However, at the wet site, the increase in tracheid lumen area and the decrease in wood density were more pronounced than at the dry site. We hypothesize that, at the dry site, environmental conditions determined by rainfall events in autumn were not sufficiently favourable to promote such a marked increase in wood growth as observed at the wet site. Indeed, for the wet site, the climatic correlations showed a positive relationship between the tracheid lumen area measured in the IADF region and precipitation in autumn (October-December), which agrees with the results by [45]. Autumn precipitation influenced cell differentiation by increasing the turgor of xylem cells during the enlargement phase, consequently producing earlywood-like cells within latewood [62]. In trees growing at the dry site, such a correlation was not found. On the contrary, we found a positive correlation between earlywood tracheid lumen area and maximum temperature in late spring. At the wet site, wood density seemed not to be affected by precipitation, but was more sensitive to maximum temperature from January to March. At the dry site, conversely, precipitation from January to March positively influenced earlywood density combined to the increase in temperature in the previous autumn (October–December).

The higher water availability at the wet site likely promoted the formation of tracheids with a larger lumen in the second peak of growth. The formation of such tracheids with wide lumen seemed to be enhanced by the occurrence of warm/dry conditions during most of the growing season, in particular from March to May, followed by a wet October. The stabilized frequency of IADFs at the wet site was negatively correlated with the precipitation of the previous November, in agreement with similar relations found also by [46] in *P. pinaster* growing at the central coast in Portugal, where IADF frequency was linked to low precipitation in November of the previous year. In this respect, scarce winter precipitation likely reduces the moisture availability during the subsequent growing season, increasing water stress, which is particularly noticeable during the summer, slowing down cambial activity and causing an earlier start in latewood formation [46]: this would increase the probability of IADF formation in case of rain events during longer durations of latewood formations. High precipitation in the current summer and autumn enhanced IADF occurrence [44–46,63]. Given that water availability does not seem to be a limiting factor at the wet site, IADF formation could be triggered by warm conditions in September combined with the peak in rain events in autumn. Such finding is in agreement with those of a previous study on *Arbutus unedo* growing at a mesic and a xeric site on Elba Island, in which the importance of mild temperature in September was already reported, associated with higher water availability for the plant to form *L*-IADFs at the mesic site [19]. At the xeric site, the second peak of the cell lumen area was not so pronounced as at the mesic site, likely due to the lower water availability.

Stabilized IADF frequency at dry site was positively correlated with maximum temperature in December. Such a relationship suggests the hypothesis of a prolonged cambial activity, lasting in December, with the formation of *L*-IADFs after autumn favorable conditions, as already reported in *P. halepensis* and *P. pinea* from Mediterranean sites [35,64].

### 4.2. IADF Formation in Q. ilex

In regards to the *Q. ilex* series of vessel lumen areas, they did not show a clear bimodal pattern, although *L*-IADFs were found at the end of the tree-ring. Recently, xylogenesis investigations showed

that in *Q.ilex* the highest number of cambial cells is produced in spring, and a second slight reactivation of cambial cells is registered after the first rainfall at the end of August [65].

We found a clear increase in vessel lumen area at the end of the ring only in rings with IADFs at the dry site, following the gradual decrease of values from earlywood to latewood. This increase in vessel lumen area was accompanied by a decrease in vessel frequency, while wood density seemed to maintain constant values. The increase in vessel lumen area at the end of the ring seemed to be promoted by wet and warm conditions from April to June (the period in which most of the tree-ring may be formed in this species) [27,28,38,66], which could favor the resumption of cambial activity in autumn leading to the IADF formation. At the dry site, the positive and negative correlations found between vessel lumen area and maximum temperature and precipitation of April–June, respectively, may be also considered a consequence of improved carbohydrate availability due to better photosynthetic performance which would favour cell wall formation during xylogenesis, thus constraining cell expansion; this would change the relative rates of the various processes during cell enlargement and differentiation, which are at the base of the transition between earlywood and latewood [67]. Wet and warm conditions from April to June also seemed to have a negative influence on vessel frequency, while density at the end of the ring was negatively correlated with winter temperature. Wood density at the beginning of latewood seemed to be negatively related to precipitation from April to June, while the increase in temperature in these months positively influenced vessel frequency. Such results agree with [38], which also found a positive correlation between vessel size and spring precipitation and a negative correlation between density and precipitation in early summer. Precipitation during the summer could be related to the formation of new leaves in *Q. ilex* with the deposition of an extra growth band of xylem [18,27,28,33,36]. Correlations between climatic data and stabilized IADF frequency at the dry site were not significant, which suggests that the formation of IADFs under dry conditions could be triggered by different interconnected factors from those acting under wet conditions.

At the wet site, in *Q. ilex*, and in *P. pinaster*, conduit lumen area and wood density at the end of the ring were positively affected by autumn precipitations, which also influenced negatively vessel frequency. The higher amount of water availability, in this case, seemed to induce the formation of more efficient wood, with larger vessel lumens and lower vessel frequencies in comparison to the dry site. The stabilized IADF frequency proved to be negatively correlated with the maximum temperature of the previous September and current November, which can affect cambial activity at the beginning and the end of the vegetative period.

The different influences of climatic variables on different sectors of the tree-ring can be explained by the fact that environmental parameters exert their action at the same moment, but on cells that are in different stages of development, thus involving different metabolic processes. Therefore, they likely have different sensitivities to temperature and water availability and can react in different ways.

## 5. Conclusions

Both species formed a similar type of IADFs, namely the *L*-IADFs, which could increase the hydraulic conductivity late in the growing season or increase the water storage capacity of trees due to their larger lumen area [46]. Indeed, tracheids of latewood IADFs are earlywood-like cells, thus they have larger diameters and higher hydraulic efficiency than "true" latewood cells [68,69].

We found a few significant correlations between IADF features and climatic factors, and they were not always the same in the two species and at the two sites, confirming that the factors triggering IADF formation are site- and species-specific. This is in agreement with findings on other species at other Mediterranean sites. However, similarities in the response of the two species to the environmental factors were more marked at the dry site than wet site, indicating a climatic constraint determined by arid conditions [9].

We showed that in the two analysed species at both sites, specific wood anatomical traits are more useful proxies than IADF frequency to encode different climate signals recorded in tree-rings. Moreover, the degree of plasticity of these traits to the seasonal climatic conditions, as well as IADF

frequency, is surely controlled by intrinsic factors. Therefore, complex relations among intrinsic and environmental factors may mask straightforward relations between IADF features and water availability or temperature.

The use of IADFs in dendroecology can improve the resolution of the climate signal within the growing season [62], obtaining information about the physiological and ecological significance of IADFs as adaptive traits [45], especially when combined with fine quantitative wood anatomy and xylogenesis analyses, because the different wood traits are differentially controlled by environmental factors during xylogenesis.

**Author Contributions:** Conceptualization, A.B., P.C. and V.D.M.; Data curation, A.B., G.B. and V.D.M.; Formal analysis, A.B. and G.B.; Funding acquisition, P.C.; Investigation, A.B., G.B. and V.D.M.; Methodology, A.B. and V.D.B.; Supervision, V.D.M.; Writing – original draft, A.B. and V.D.M.; Writing – review and editing, A.B., G.B., P.C. and V.D.M. All authors have read and agreed to the published version of the manuscript.

**Funding:** This research received no external funding.

**Acknowledgments:** The authors wish to thank: L. Nardella (Parco Nazionale dell'Arcipelago Toscano) and D. Giove (Comunità Montana dell'Arcipelago Toscano) and M. Nötzli for assistance in the field; W. Schoch, H. Gärtner and G. Aronne for helpful support and suggestions during the laboratory phase of this project. This research is linked to activities conducted within the COST FP1106 'STReESS' network.

**Conflicts of Interest:** The authors declare no conflict of interest.

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
