# Peer review of "Xylem Plasticity in Pinus pinaster and Quercus ilex Growing at Sites with Different Water Availability in the Mediterranean Region: Relations between Intra-Annual Density Fluctuations and Environmental Conditions"

_forests, doi:10.3390/f11040379_

Round 1

Reviewer 1 Report

Excellent  paper. Well described methods, research design  and results.

Author Response

We thank this reviewer for appreciating our study.

Reviewer 2 Report

A title with abbreviations is not advisable. I suggest:

"Xylem plasticity in Pinus pinaster and Quercus ilex growing in the Mediterranean region: relations between intra-annual density fluctuations and environmental conditions".

I would remove from the title the reference to the sites because both sampling areas are defined, in terms of climate, by the same meteorological station, which is about 400 m lower than the area of study (is there any altitude correction of the temperature and precipitation data?). In fact, the only evidence of differences between both sites is the presence of "more mesic species" (line 105) and this is not a strong difference to justify the site as a source of variation in the results.

Other methodological features are weak, as the absence of explanation for some of the measuring methods and for dome of the abbreviations (GLK, rbar), the unsuitable use of the term “wood density” for the ratio area occupied by cell walls over total area, the number of samples trees (10 trees per species and site or 5 trees per species and site?).

The manuscript offers results by means of figures instead of tables, but I think some table or tables with the descriptive statistics of the results for the main variables involved is needed. The figures are difficult to understand due to several reasons:

    figure 2 and 4: caption refers to parts of the figure not visible in the figure itself

    figure 3: vertical axes should present the same scale; "sample depth" is undefined

    figure 4: "MS" and "XS" are not defined

    figure 5: scale in vertical axes is not appropriate for seeing differences between columns and errors; confusion between sources of variation and significance letters (a, b, c, etc.); undefined values in horizontal axes

    figures 7 and 8: white asterisks undefined; mixture of capital and lowercase letters in horizontal axes; values 0,4, -0,4… undefined

    figure 9: caption must be clarified 

Please use the term "vessel" just for hardwoods and the term "tracheid" just for coniferous trees.

In chapter 3.3. it is expected to find some values of correlation coefficients.

Author Response

Responses to Reviewer n. 2

C1

A title with abbreviations is not advisable. I suggest: "Xylem plasticity in Pinus pinaster and Quercus ilex growing in the Mediterranean region: relations between intra-annual density fluctuations and environmental conditions".

R1

We modified the title also according to the suggestion of another reviewer.

C2

I would remove from the title the reference to the sites because both sampling areas are defined, in terms of climate, by the same meteorological station, which is about 400 m lower than the area of study (is there any altitude correction of the temperature and precipitation data?). In fact, the only evidence of differences between both sites is the presence of "more mesic species" (line 105) and this is not a strong difference to justify the site as a source of variation in the results.

R2

We prefer keeping the reference to the two sites in the title. About the description of the sites, we refer to Battipaglia et al. 2010 in which a detailed description of the site is reported, including water availability in the soil. We added info about soil water holding capacity of the two sites. Lines 118-120.

C3

Other methodological features are weak, as the absence of explanation for some of the measuring methods and for dome of the abbreviations (GLK, rbar), the unsuitable use of the term “wood density” for the ratio area occupied by cell walls over total area, the number of samples trees (10 trees per species and site or 5 trees per species and site?).

R3

We added requested details and clarifications in the methods. Lines 138-141.

C4

The manuscript offers results by means of figures instead of tables, but I think some table or tables with the descriptive statistics of the results for the main variables involved is needed. The figures are difficult to understand due to several reasons:

    figure 2 and 4: caption refers to parts of the figure not visible in the figure itself

    figure 3: vertical axes should present the same scale; "sample depth" is undefined

    figure 4: "MS" and "XS" are not defined

    figure 5: scale in vertical axes is not appropriate for seeing differences between columns and errors; confusion between sources of variation and significance letters (a, b, c, etc.); undefined values in horizontal axes

    figures 7 and 8: white asterisks undefined; mixture of capital and lowercase letters in horizontal axes; values 0,4, -0,4… undefined

R4

Figures from n.3 to n.9 were improved by modifying them and adding clarifications as requested. Captions were amended; MS and XS were replaced by wet and dry; the scale in the y axis of fig. 5 should remain 100 as maximum because it is a percent; the values in the horizontal axis are the number (from 1 to 5) of the regions in which the ring is divided. There is no reason to keep white and black asterisks, so we converted all of them in black.

Sample depth represents the number of samples for each year. This was specified in the legend of figure 3.

C5

figure 9: caption must be clarified

R5

We corrected the legend.

C6

Please use the term "vessel" just for hardwoods and the term "tracheid" just for coniferous trees.

R6

Of course, sorry, if it was mispelled somewhere. We checked throughout the text.

C7

In chapter 3.3. it is expected to find some values of correlation coefficients.

R7

We preferred not showing the values in the text for not reducing readability of a text that is already dense of results.

Reviewer 3 Report

Comments

This manuscript submitted by Angela Balzano1 et al. addresses xylem plasticity of Pinus pinaster and Quercus ilex growing at two different sites (Wet and Dry) on the Elba island in Central Italy, based on dendrological and wood anatomical analysis techniques and points.  

I think that the main findings of this manuscript will be important for the full understanding of the xylem plasticity of many species and sites to climatic variation, especially in the Mediterranean region. However, in the present state, I cannot recommend this manuscript because it has extensive problems in text and figures and the reliability of this manuscript is very low. It was very hard for me to understand the results and discussions. I recommend to the authors for re-writing this manuscript.

Major points

Point 1

The authors demonstrate their findings by making compare between the response of wet site and dry site. However, the analysis is based only precipitation and maximum temperature data from one site that is 10 km-far from both sites and lower altitude. In addition, the evidence related to the condition (dry or wet) of each site did not indicate in the manuscript. Only description is xeric species and mesic species and the ecological background of the existence of such species in each site is also unclear. Please provide more information of each site (for example, soil moisture content, meteorological data) to improve reliability of your work.

Point 2

I could not understand and evaluate properly the results properly and discussion, because there are many mistakes in your figures. Please revise all figures including legends and the related text.

Minor points 

Title

L3. “IADFs” should be change to “intra-annual density fluctuations”, because it would be not general term for most of readers in other fields.

Abstract

L18. Suggest you change “Wood density” to “wood density measured as the percentage of cell walls over total xylem area”

Introduction

L77. Delete a space between Ref 38 and 39

Materials and Methods

L105. Delete 0 after 46'

L120. Explain “LINTAB system” in detail or cite references

L144-148, “for p. pinaster...”. Put this paragraph before the paragraph (L139-143)

L169-174. You partitioned growth ring into 4 regions and describe earlywood and latewood in results and discussion. Explain the relation between each region and earlywood or latewood, and indicate how do you define earlywood and latewood.

In addition, please provide data on average and SD of ring width for each species and information on xylogenesis patter (in general, when it starts and ends in case of P. pinaster and Q. ilex).

L204, Figure 3a, It should be “3b”

L205, “Dry”.  Wet?

L206, “Wed”.  Dry?

L220-228, “In P. pinaster...”. Since Figure 4 is incorrect, this section or figure 4 should be revised.

L223, “70%”. Is it true?

L236, “Dry site”. Is it true?

L237, “90%”. Is it true?

L242-245, “As in P. pinaster, ...”. I suggest putting a paragraph break.

L245-248, “Vessel frequency ...”. I suggest putting a paragraph break.

L267-314. In this section, you explain that region 4 is latewood (L285, L289, L291, L295 etc.). As mentioned above, I suggest explaining the relation between each region and latewood at the materials and methods.

L289-290. “At the Dry site, …”. This result is interesting for me. Please give me your comment on why these correlations are obtained.

L292, “November”. It should be “December”.

L292, L293, “8d”. It should be “8f”

L293, “At the Wet site …”  Please explain why these results are obtained.

L296, “8f”. It should be “8g”

L298, “8g”. It should be “8h”

L299, “8h”. It should be “8i”

L298-300, “June to September”. It should be “April to Jun”

L301, L303, “8i”. It should be “8l”

L305, “8l”. It should be “8m”

L306, “8m”. It should be “8n”

L311-313, “Wet site”. Figure 9d indicate dry site.

Discussion

L353, “lumen area tracheids”. Change to “tracheids lumen area”.

L358, “vessle lumen area”. Correct to “tracheid lumen area”

L422

Figures

Figure 2. Caption is incorrect.

Figure 4. I suggest putting legend (a)-(d) in each figure. Change “MS” and “XS” to “Wet” and “Dry”. Switch the figure for P. pinaster to the figure for Q. ilex.

Figure 5. I suggest putting legend (a)-(h) and “Dry”, “Wet” in each figure.

Figure 6. I suggest putting legend (a)-(d) and “Dry”, “Wet” in each figure.

Figure 7. I suggest putting legend “Dry”, “Wet” in each figure. It should be “P<0.05”, not “P 0.05”

It should be “October”, not “September”. Please use black asterisks, not white asterisks. Put a space “of the”

Figure 8. Captions is described differently in the text. Please make these consistent

Figure 9. Caption of Figure d is described differently in the text. Please make these consistent.

References

L460-462  IAWA 40, 2, 155-182

L487  Jalkanen

L521  Battipaglia

I will appreciate it if you accept all suggested corrections.

Author Response

Responses to Reviewer n. 3

C2

This manuscript submitted by Angela Balzano1 et al. addresses xylem plasticity of Pinus pinaster and Quercus ilex growing at two different sites (Wet and Dry) on the Elba island in Central Italy, based on dendrological and wood anatomical analysis techniques and points.  

I think that the main findings of this manuscript will be important for the full understanding of the xylem plasticity of many species and sites to climatic variation, especially in the Mediterranean region. However, in the present state, I cannot recommend this manuscript because it has extensive problems in text and figures and the reliability of this manuscript is very low. It was very hard for me to understand the results and discussions. I recommend to the authors for re-writing this manuscript.

R2

We are grateful to this reviewer for having highlighted the unclear points giving us the possibility to considerably improve the manuscript.

Point 1

C3

The authors demonstrate their findings by making compare between the response of wet site and dry site. However, the analysis is based only precipitation and maximum temperature data from one site that is 10 km-far from both sites and lower altitude. In addition, the evidence related to the condition (dry or wet) of each site did not indicate in the manuscript. Only description is xeric species and mesic species and the ecological background of the existence of such species in each site is also unclear. Please provide more information of each site (for example, soil moisture content, meteorological data) to improve reliability of your work.

R3

Detailed description of the site is given in Battipaglia et. al 2010 where we explained also the origin of the climate data. We referred to this paper in the manuscript. However, we added information of soil water holding capacity. Lines 118-122.

C4

I could not understand and evaluate properly the results properly and discussion, because there are many mistakes in your figures. Please revise all figures including legends and the related text.

R4

We revised figures from n.3 to n.9 and related text and legends.

C5

L3. “IADFs” should be change to “intra-annual density fluctuations”, because it would be not general term for most of readers in other fields.

R5

Title has been changed as requested.

C6

L18. Suggest you change “Wood density” to “wood density measured as the percentage of cell walls over total xylem area”

R6

We added this specification in the abstract. Lines 31-32.

C7

L77. Delete a space between Ref 38 and 39; L105. Delete 0 after 46'

R7

Done

C8

L120. Explain “LINTAB system” in detail or cite references

R8

We added the citation. Lines 135-136

C9

L144-148, “for p. pinaster...”. Put this paragraph before the paragraph (L139-143)

R9

We moved the paragraph as requested. Lines 160-172

C10

L169-174. You partitioned growth ring into 4 regions and describe earlywood and latewood in results and discussion. Explain the relation between each region and earlywood or latewood, and indicate how do you define earlywood and latewood.

R10

In all the tree rings the division was standardized and each ring divided into four or five equal parts depending on presence or absence of IADFs, as reported in method. The distinction between earlywood and latewood was based on the application of Mork’s definition (Mork, 1928). Details have been added in the introduction and methods. Lines 61-63; 200-205; 219-220.

C11

In addition, please provide data on average and SD of ring width for each species and information on xylogenesis patter (in general, when it starts and ends in case of P. pinaster and Q. ilex).

R11

Data about TRW have been added in the text at lines: 238-244.

  1. pinaster in Mediterranean site showed that in summer and winter the number of cambial cells in the cambium reach the minimum (Vieira et al 2015). Q. ilex reaches the maximum peak of cambial activity in spring, and a small peak of cambial activity is recorded after rainfall in the end of August (Montagnoli et al 2019). We added these information and related references in the text. Lines 402-404; 453-456.

C12

L204, Figure 3a, It should be “3b”; L205, “Dry”.  Wet?; L206, “Wed”.  Dry?

R12

We corrected all these typoes.

C13

L220-228, “In P. pinaster...”. Since Figure 4 is incorrect, this section or figure 4 should be revised.

R13

We modified and corrected the figure. Q.ilex and Pinus were inverted; we replaced MS and XE with  Wet and Dry. Further, we added lettering.

C14

L223, “70%”. Is it true? L236, “Dry site”. Is it true? L237, “90%”. Is it true?

R14

We checked and these affirmations were correct.

C15

L242-245, “As in P. pinaster, ...”. I suggest putting a paragraph break; L245-248, “Vessel frequency ...”. I suggest putting a paragraph break.

R16

Done

C16

L267-314. In this section, you explain that region 4 is latewood (L285, L289, L291, L295 etc.). As mentioned above, I suggest explaining the relation between each region and latewood at the materials and methods.

R16

We added clarifications in the introduction and methods. Lines 61-63; 200-205; 219-220.

C17

L289-290. “At the Dry site, …”. This result is interesting for me. Please give me your comment on why these correlations are obtained.

R17

We added an explanation in the discussion at lines 464-469.

C18

L292, “November”. It should be “December”.

R18

Done

C19

L292, L293, “8d”. It should be “8f”

R19

It was replaced with 8e

C20

L293, “At the Wet site …”  Please explain why these results are obtained.

R20

We added an explanation in the end of the discussion. Lines 488-492.

C21

L296, “8f”. It should be “8g”; L298, “8g”. It should be “8h”; L299, “8h”. It should be “8i”.

R21

Done

C22

L298-300, “June to September”. It should be “April to Jun”

R22

We adjusted the asterisk in the Figure 8i.

C23

L301, L303, “8i”. It should be “8l”; L305, “8l”. It should be “8m”; L306, “8m”. It should be “8n”

R23

Done

C24

L311-313, “Wet site”. Figure 9d indicate dry site.

R24

We checked and corrected

C25

L353, “lumen area tracheids”. Change to “tracheids lumen area”; L358, “vessle lumen area”. Correct to “tracheid lumen area”

R25

Done

C26

Figure 2. Caption is incorrect.

R26

We corrected the caption

C27

Figure 4. I suggest putting legend (a)-(d) in each figure. Change “MS” and “XS” to “Wet” and “Dry”. Switch the figure for P. pinaster to the figure for Q. ilex.

R27

We accepted all comments and made the corrections.

C28

Figure 5. I suggest putting legend (a)-(h) and “Dry”, “Wet” in each figure.

R28

We added letters and legends as requested.

C29

Figure 6. I suggest putting legend (a)-(d) and “Dry”, “Wet” in each figure.

R29

We added letters and legends as requested.

C30

Figure 7. I suggest putting legend “Dry”, “Wet” in each figure. It should be “P<0.05”, not “P 0.05”

R30

We modified the figure as requested

C31

It should be “October”, not “September”. Please use black asterisks, not white asterisks. Put a space “of the”

R31

We modified the figure as requested

C32

Figure 8. Captions is described differently in the text. Please make these consistent

Figure 9. Caption of Figure d is described differently in the text. Please make these consistent.

R32

We modified the text to be consistent.

C33

L460-462  IAWA 40, 2, 155-182; L487  Jalkanen; L521  Battipaglia

R33

All typoes were amended.

C34

I will appreciate it if you accept all suggested corrections.

R34

We accepted all your suggestions.

Reviewer 4 Report

Overall

This study evaluates whether variation in wood anatomical traits in two species measured at both wet and dry sites is correlated with climate. Clearly the authors spent many hours on the anatomy and dendrochronology, and there is a strong rationale for such a study. However, I identified several weaknesses in the paper. There were many grammatical mistakes. Please go through the ms again carefully because I stopped pointing them out after the introduction. The statistical analyses are not well explained and I’m not sure if they are justified. An ANOVA seems like an odd choice for non-independent observations, but perhaps I’ve just missed something. The figures and figure captions need significant edits. Abbreviations are never explained, and the text is really small. Figure 5 is of low quality. It could easily be done in R. I disagree with the conclusion that IADFs record some climate signal, because there were only a few significant correlations with climate. Again, maybe I’m missing something but I don’t see how they can come to this conclusion based on the correlation matrices.

Line by line edits

Line 22: growing at sites that varied in water availability…

Line 25: has been linked to rain events following periods of summer drought.

Line 26: increased

Line 28: Species expressed greater similarity under arid environments, as unfavorable climates constrained trait variation. 

Line 37: during the growing season

Line 38: consequence of ongoing

Line 39: increase in the frequency

Line 40: and heatwaves

Line 41-44: This sentence has awkward wording. “Increased efficiency of plant hydraulic systems is fundamental for plant survival within Mediterranean climates. In these environments, plants must be to efficiently transport water to their tissues when it is available (…), while maintaining a low propensity for embolisms (…) when water is scarce.”

Line 50: vague. Changing the functionality how? Can you be more specific? Similar comment for line 48. It’s unclear how the functionality would change. Perhaps you mean reduced performance?

Line 52: vague. Considered with regards to what? There is no information about the reference study.

Line 53: unexpected? I would remove this term. Unexpected is based on what people would think? 

Line 57: briefly explain what earlywood is versus latewood. I assume it means wood produced earlier in time or earlier in the season.

Line 63: “Given that wood anatomy…” The word ‘since’ should only be used to demarcate passage of time.

Line 66: this sentence is incomplete and grammatically incorrect. See following

Line 66: Indeed, to understand the ecological function of IADFs as an acclimation response to climate requires studies of intra-annual xylem plasticity across a range of species growing in a range of environmental conditions. This information would allow to improve…

Line 69: help obtain

Lines 73-78: not needed. Can be moved to the discussion or possibly the methods.

Line 79: combine this short paragraph with the following.

Line 85: prone to forming IADFs

Line 114: the two cores are non-independent

Line 115-116: unclear. Are you suggesting you had a cap on the number of specimens you could sample because they were protected?

Line 149: these are low sample sizes (although obviously time intensive) therefore I suggest nonparametric methods.

Line 152: not enough information about the software

Line 154: early and latewood is defined here. Move earlier.

Line 186: I am confused about the choice of one-way ANOVA for examining temporal trends. By intra-annual you mean within a year, but what are the time periods? This is confusing. Perhaps you just mean that the IADFs represent intra-annual variation. Wouldn’t you prefer a mixed model rather than ANOVA? There are also more sophisticated methods like ARIMA for temporal trends. In short, there needs to be more detail to justify the statistical analyses. Also, the cores used to estimate frequency of IADFs were non-independent, how did you account for this? You could just average them, or you could include random effect of individual tree.

Line 203: missing space after “from:

Figure 3: the axes are really hard to read. Please make the font a bit larger. What does the horizontal line present?  What does RWI represent? Ring width increment I guess

Figure 4: I am confused about the horizontal axis. What is the progressive number? Your captions are not clear. What are MS and XS?

Line 223: the 70% of what? You explain progressive number above but not in layman’s terms. Could you possibly say that progressive number is associated with depth?

Figure 5: what does the horizontal axis represent?

Figure 6: what is p/0.05? Do you mean <?

Line 284: this paragraph can be shortened and the general patterns summarized for the readers. It is very unusual to go into this much detail about a correlation matrix. Perhaps you could split this paragraph into two paragraphs to discuss precipitation versus temperature effects. 

Figure 7: it’s a bit strange that the darkest purple color is not statistically significant. I would double check this. Also, the months are sometimes capitalized and sometimes not, which is confusing at first glance. Instead, you can indicate the year above (add text) or with bold or some other color. Or perhaps just explain better in the caption.

Figure 9: I think you mean correlation coefficients on the vertical axis, and not just coefficients.

Discussion: why are wet and dry capitalized?

Line 371: This sentence is vague. Can you offer an explanation for this difference? Can you discuss again when precipitation typically falls within Mediterranean climates? And perhaps relate your findings back to what these species typically experience? I see that you explain in the following paragraph. This is  a  bit strange because I expect the next paragraph to being a new idea, so perhaps you just want to move things around a bit or even just give a brief explanation and then go into detail in the next paragraph.

Line 434: avoid the word “proved”

Conclusions: I was very surprised by how few climate-growth and climate-IADFs correlations were significant. Does this not indicate that IADFs are weakly controlled by the chosen climate variables? Perhaps there are other climate variables you should consider? Or perhaps it’s more about soil water availability than precipitation per se? so really, I disagree with the conclusion that IADFs record climate signals.

Author Response

Responses to Reviewer n. 4

C35

This study evaluates whether variation in wood anatomical traits in two species measured at both wet and dry sites is correlated with climate. Clearly the authors spent many hours on the anatomy and dendrochronology, and there is a strong rationale for such a study. However, I identified several weaknesses in the paper. There were many grammatical mistakes. Please go through the ms again carefully because I stopped pointing them out after the introduction. The statistical analyses are not well explained and I’m not sure if they are justified. An ANOVA seems like an odd choice for non-independent observations, but perhaps I’ve just missed something. The figures and figure captions need significant edits. Abbreviations are never explained, and the text is really small. Figure 5 is of low quality. It could easily be done in R. I disagree with the conclusion that IADFs record some climate signal, because there were only a few significant correlations with climate. Again, maybe I’m missing something but I don’t see how they can come to this conclusion based on the correlation matrices.

R35

We are grateful to this reviewer for having highlighted the unclear points giving us the possibility to explain them better to improve the manuscript.

C36

Line by line edits

Line 22: growing at sites that varied in water availability…

Line 25: has been linked to rain events following periods of summer drought.

Line 26: increased

Line 28: Species expressed greater similarity under arid environments, as unfavorable climates constrained trait variation.

Line 37: during the growing season

Line 38: consequence of ongoing

Line 39: increase in the frequency

Line 40: and heatwaves

R36

We amended all typoes

C37

Line 41-44: This sentence has awkward wording. “Increased efficiency of plant hydraulic systems is fundamental for plant survival within Mediterranean climates. In these environments, plants must be to efficiently transport water to their tissues when it is available (…), while maintaining a low propensity for embolisms (…) when water is scarce.”

R37

We rephrased the sentence as requested. Lines 46-50

C38

Line 50: vague. Changing the functionality how? Can you be more specific? Similar comment for line 48. It’s unclear how the functionality would change. Perhaps you mean reduced performance? Earlywood and latewood have different 

R38

We amended the sentences. Lines 57-63

C39

Line 52: vague. Considered with regards to what? There is no information about the reference study.

R39

We mean that they were classified as functional wood traits (in IAWA list of wood anatomical traits as reported in the references associated). Lines 58-60.

C40

Line 53: unexpected? I would remove this term. Unexpected is based on what people would think?

R41

We removed the term.

C41

Line 57: briefly explain what earlywood is versus latewood. I assume it means wood produced earlier in time or earlier in the season.

R41

Although the term earlywood and latewood should be quite common, we added a brief description. Lines 61-63

C42

Line 63: “Given that wood anatomy…” The word ‘since’ should only be used to demarcate passage of time.

R42

Done

C43

Line 66: this sentence is incomplete and grammatically incorrect. See following

Line 66: Indeed, to understand the ecological function of IADFs as an acclimation response to climate requires studies of intra-annual xylem plasticity across a range of species growing in a range of environmental conditions. This information would allow to improve…

R43

We rephrased the sentence. Lines76-82.

C44

Line 69: help obtain

R44

Done

C45

Lines 73-78: not needed. Can be moved to the discussion or possibly the methods.

We prefer to keep this part in the Introduction in order to inform the readers about the state of art and the gaps regarding the “tracheidogram method”. This will help the readers to follow the main aims of the paper.

C46

Line 79: combine this short paragraph with the following.

R46

Done. Lines 92-95

C47

Line 85: prone to forming IADFs

R47

Done

C48

Line 114: the two cores are non-independent

R48

We specified in the text. Lines 129-130.

C49

Line 115-116: unclear. Are you suggesting you had a cap on the number of specimens you could sample because they were protected?

R49

Yes: trees were in a National Park with restrictions in sampling.

C50

Line 149: these are low sample sizes (although obviously time intensive) therefore I suggest nonparametric methods.

R50

We added in the text that the data were normally distributed, thus we could choose to use a parametric method even if sample size was relatively low.

C51

Line 152: not enough information about the software

R51

We specified that it is an image analysis software and added a few details. It is commonly used in quantitative anatomy. Lines 181-182.

C52

Line 154: early and latewood is defined here. Move earlier.

R52.

We added a short description in the Introduction. Lines 61-63

C53

Line 186: I am confused about the choice of one-way ANOVA for examining temporal trends. By intra-annual you mean within a year, but what are the time periods? This is confusing. Perhaps you just mean that the IADFs represent intra-annual variation. Wouldn’t you prefer a mixed model rather than ANOVA? There are also more sophisticated methods like ARIMA for temporal trends. In short, there needs to be more detail to justify the statistical analyses. Also, the cores used to estimate frequency of IADFs were non-independent, how did you account for this? You could just average them, or you could include random effect of individual tree.

R53

Intra annual density fluctuation (IADF) is the way in which we indicate the region of the rings where we identify density fluctuation. We used ANOVA to compare the number and the frequency of IADF identified in the different species during all the analysed period. Thus we are not performing a temporal analyses but we are testing groups to see if there’s a difference between them.

Frequency of IADFs is identified in each core, that represent and individual and independent sample. However each sample is then averaged to furnish a mean IADF frequency.

C54

Line 203: missing space after “from

R54

Done

C55

Figure 3: the axes are really hard to read. Please make the font a bit larger. What does the horizontal line present?  What does RWI represent? Ring width increment I guess

R55

We increased the font. The horizontal line represents the year associated with each tree ring, RWI represents the Ring Width Index. We added clarifications in the legend.

C56

Figure 4: I am confused about the horizontal axis. What is the progressive number? Your captions are not clear. What are MS and XS?

R56

The progressive number is the number associated to each vessels encounter along the tree ring with. We added specifications in the text. Lines 186-190. We replaced MS and XS with Wet and Dry throughout the text and figures.

C57

Line 223: the 70% of what? You explain progressive number above but not in layman’s terms. Could you possibly say that progressive number is associated with depth?

R57

70% of the vessels along the tree ring width, we specified in the text. Refer also to R56.

C58

Figure 5: what does the horizontal axis represent? 

R58

The horizontal axis represents  the progressive number, which  is the number associated to each vessels encountered along the tree ring with. Please refer to R56

C59

Figure 6: what is p/0.05? Do you mean <?

R59

Yes, we corrected

C60

Line 284: this paragraph can be shortened and the general patterns summarized for the readers. It is very unusual to go into this much detail about a correlation matrix. Perhaps you could split this paragraph into two paragraphs to discuss precipitation versus temperature effects.

R60

We believe that it is important to comment all the significant correlations since they have important ecological implications. Further, splitting the paragraph in two parts means that we should repeat several concepts and variables reducing readability of the text. The related figures are a summary of this long paragraph. Moreover, the other reviewers asked for more details about these correlations.

C61

Figure 7: it’s a bit strange that the darkest purple color is not statistically significant. I would double check this. Also, the months are sometimes capitalized and sometimes not, which is confusing at first glance. Instead, you can indicate the year above (add text) or with bold or some other color. Or perhaps just explain better in the caption.

R61

We checked all data and formatted the legends.

C62

Figure 9: I think you mean correlation coefficients on the vertical axis, and not just coefficients.

R62

Yes, we corrected

C63

Discussion: why are wet and dry capitalized?

R63

No reason, we corrected

C64

Line 371: This sentence is vague. Can you offer an explanation for this difference? Can you discuss again when precipitation typically falls within Mediterranean climates? And perhaps relate your findings back to what these species typically experience? I see that you explain in the following paragraph. This is  a  bit strange because I expect the next paragraph to being a new idea, so perhaps you just want to move things around a bit or even just give a brief explanation and then go into detail in the next paragraph.

R64

We moved the paragraph. Lines 405-407.

C65

Line 434: avoid the word “proved”

R65

We replaced “proved” with ‘’showed’’

C66

Conclusions: I was very surprised by how few climate-growth and climate-IADFs correlations were significant. Does this not indicate that IADFs are weakly controlled by the chosen climate variables? Perhaps there are other climate variables you should consider? Or perhaps it’s more about soil water availability than precipitation per se? so really, I disagree with the conclusion that IADFs record climate signals.

R66

We smoothed our conclusion, because of course other internal or external variables could control IADF, but the tested factors were shown to influence IADF formation. To get more precise correlations, we should know the period in which the cells were produced. For this reason, we suggest to investigate the xylogenesis.

Round 2

Reviewer 2 Report

I would remove from the title the reference to the sites because both sampling areas are defined, in terms of climate, by the same meteorological station, which is about 400 m lower than the area of study (is there any altitude correction of the temperature and precipitation data?). In fact, the only evidence of differences between both sites is the presence of "more mesic species" (line 105) and this is not a strong difference to justify the site as a source of variation in the results.

The manuscript offers results by means of figures instead of tables, but I think some table or tables with the descriptive statistics of the results for the main variables involved is needed.

figure 3: vertical axes should present the same scale in figures at both sides.

figure 5: scale in vertical axes is not appropriate for seeing differences between columns and errors (I suggest a scale from 80 % to 100 %); undefined values in caption for horizontal axes.

figures 7 and 8: 0,4; ...-0,4 undefined (at right side)

depth represents the number of samples for each year. This was specified in the legend of figure 3.

In chapter 3.3. it is expected to find some values of correlation coefficients, because they are the topic involved.

Author Response

C1

I would remove from the title the reference to the sites because both sampling areas are defined, in terms of climate, by the same meteorological station, which is about 400 m lower than the area of study (is there any altitude correction of the temperature and precipitation data?). In fact, the only evidence of differences between both sites is the presence of "more mesic species" (line 105) and this is not a strong difference to justify the site as a source of variation in the results.

R1

Done. We replaced “xeric and mesic sites” with “sites with different water availability”. Please, find here the reference: https://nph.onlinelibrary.wiley.com/doi/pdf/10.1111/j.1469-8137.2010.03443.x

C2

The manuscript offers results by means of figures instead of tables, but I think some table or tables with the descriptive statistics of the results for the main variables involved is needed.

R2

Descriptive statistics in terms of mean values and standard errors of all analysed parameters are already reported in the graphs. Since it is not advisable to report the same data both in graphs and tables, we prefer to keep the original output as graphs in the manuscript. We believe that reporting the data in graphs in this specific case gives more the idea of how the parameters vary along the ring width. To make this “spatial variability” within the ring width clearer, we modified the graphs: we indicated that the X-axis is oriented as the tree ring by adding a graphical information about the boundaries of earlywood and latewood (ref. Figure 4, 5, 6).

C3

Figure 3: vertical axes should present the same scale in figures at both sides.

R3

We redraw the figures to make them clearer.

C4

Figure 5: scale in vertical axes is not appropriate for seeing differences between columns and errors (I suggest a scale from 80 % to 100 %); undefined values in caption for horizontal axes.

R4

Being percentage values, we cannot modify the Y axis. However, we found a graphical solution to better show the differences. Moreover, please consider that letters with significance of ANOVA are also reported.

The values in the horizontal axis are the numbers of the analysed regions within each tree ring. We understand that the number can be misleading and gives the impression that we forgot to change the labels, so we added the letter R to indicate R1 = Region 1; R2 = Region 2.... in Fig. 5 and Fig. 6.

C5

figures 7 and 8: 0,4; ...-0,4 undefined (at right side)

R5

We changed Figs 7 and 8 specifying it is the scale for the correlation coefficient.

C6

depth represents the number of samples for each year.

R6

Yes, we specified this in the legend of figure 3.

C7

In chapter 3.3. it is expected to find some values of correlation coefficients, because they are the topic involved.

R7

We addedd the values in the text.

Reviewer 3 Report

The manuscript has been revised well. 

Author Response

We thank the reviewer.

Reviewer 4 Report

Line 41: to pronounced drought stress

Line 49: While maintaining a low propensity for embolisms

Line 60: hydraulic system

Line 61: as a result of

Line 63: hydraulic conductivities?

Line 67: linked with

Line 69: stomatal closure

Line 74: and the ability

Line 78: would allow us

>Stopped editing the English at this point, too many<

Line 221: samples normality? Do you mean assumptions of normality?

This section regarding statistical analyses is still unclear. In the response, they say that they used ANOVA because they just have the number and frequency (how is this different from number?) of the IADFs. It also still doesn’t say that the cores were averaged, but they said yes in the response to reviewers.

Please note that I still disagree with the conclusions. You show only a few significant correlations with climate, and although in the response you say that you smoothed the conclusion, it reads the same way.

Author Response

C1

Line 41: to pronounced drought stress Done

Line 49: While maintaining a low propensity for embolisms Done

Line 60: hydraulic system Done

Line 61: as a result of Done

Line 63: hydraulic conductivities? Done

Line 67: linked with Done

Line 69: stomatal closure Done

Line 74: and the ability Done

Line 78: would allow us Done

>Stopped editing the English at this point, too many<

R1

We corrected all typoes and revised the English again throughout the text.

C2

Line 221: samples normality? Do you mean assumptions of normality?

R2

Yes. We added specification in the text al lines 208-215.

C3

This section regarding statistical analyses is still unclear. In the response, they say that they used ANOVA because they just have the number and frequency (how is this different from number?) of the IADFs. It also still doesn’t say that the cores were averaged, but they said yes in the response to reviewers.

R3

Regarding the IADFs, the number of them was counted and used to calculate the frequency following the standard methodologies commonly accepted in literature. Then IADF frequency as well as all anatomical parameters of each species at the two sites were processed with oneway ANOVA, using Student–Newman–Keuls coefficient for multiple comparison tests (p<0.05). The SPSS statistical package was used (SPSS Inc.; Chicago, IL, USA). Lines 208-215.

Regarding the cores, we added this information in the text. The chronologies obtained from individual cores where averaged. Lines 123-124.

C4

Please note that I still disagree with the conclusions. You show only a few significant correlations with climate, and although in the response you say that you smoothed the conclusion, it reads the same way.

R4

We changed the conclusion.